# Temperature Dependence of the Kinetic Parameters of the Titanium–Magnesium Catalyzed Propylene Polymerization

**DOI:** 10.3390/polym14235183

**Published:** 2022-11-28

**Authors:** Veronika Bronskaya, Galiya Manuyko, Guzel Aminova, Olga Kharitonova, Denis Balzamov, Alsu Lubnina

**Affiliations:** 1Institute of Mechanical Engineering for Chemical and Petrochemical Industry, Kazan National Research Technological University, 420015 Kazan, Russia; 2Institute of Computational Mathematics and Information Technologies, Kazan Federal University, 420008 Kazan, Russia; 3Department of Energy Supply of Enterprises, Construction of Buildings and Facilities, Kazan State Power Engineering University, 420066 Kazan, Russia

**Keywords:** propylene polymerization, heterogeneous titanium–magnesium catalysts, heterogeneity of active centers, identification of kinetic parameters, activation energy of the reaction

## Abstract

This paper provides a study of the liquid-phase polypropylene polymerization on a heterogeneous titanium–magnesium Ziegler–Natta-type catalyst. A kinetic model was developed that included the activation of potential active centers, chain growth, transferring the chains to hydrogen and monomer, and the deactivation of active centers. The model was created to predict the polymerization rate, polymer yield, and average molecular weights of polymer chains where the polymerization temperature changes from 40 to 90 °C. In developing polycentric kinetic models, there is a difficulty associated with evaluating the kinetic constants of the rates of elementary reactions/stages in polymerization. Each heterogeneous titanium–magnesium catalyst (TMC), including a co-catalyst, as well as an internal and an external electron donor, has its own set of kinetic parameters. Therefore, its kinetic parameters must be defined for each new catalyst. The presented algorithm for identifying the kinetic constants of rates starts with a kinetic model that considers one type of active centers. At the second stage, a deconvolutional analysis is used for the molecular weight distribution (MWD) of the gel permeation chromatography (GPC) data of the polypropylene samples and the most probable distribution of Flory chain lengths is found for each type of active centers. At the third stage, the single-center model is transformed into a polycentric kinetic model. For the catalyst system, five types of active centers were identified, together with a mass fraction and a number-average molecular weight for each active center type of the catalyst, which is consistent with the published results for similar Ti-based Ziegler–Natta catalysts.

## 1. Introduction

One of the most widely used and produced plastics is polypropylene (PP). PP grades are different due to the choice of a catalytic system and the polymerization conditions.

Spheripol technology is widely used in the industrial production of PP. The technology is implemented in a pre-polymerization reactor (T = 40–60 °C) and two tubular loop reactors of the main polymerization (T = 65–80 °C), which are connected in series. A reaction mixture, driven by axial pumps, circulates in the reactors at a high speed with a high recirculation ratio. The temperature in the reactor affects the polymerization speed and the properties of the polymer, therefore it must be adjusted.

Introducing new catalytic systems into production requires the creation of mathematical models in order to debug the polymerization process and predict the properties of the resulting polymer [1,2]. Studying the influence of polymerization conditions, such as the catalytic system composition, monomer pressure, and temperature on the concentration of active centers and on the chain growth constant helps to establish the mechanisms of the forming and transforming of the active centers of the catalyst, which provides the basis for synthesizing the PP grades with predefining properties.

The literature data analysis showed that the process activity at the initial polymerization stage was determined by forming many potential active centers (AC) as a result of the catalyst fragmentation. The polymerization temperature affects both the polymerization activity and the formation and distribution of highly irregular ACs during PP polymerization. At the optimal polymerization temperature, adding a small amount of hydrogen to the polymerization system significantly increases the percentage of activated centers from inactive centers on the catalyst surface. It is difficult to accurately quantify the active centers of metal (Ti) per mass of the activated catalyst; it is often assumed that all Ti atoms that are present in the catalyst mass participate in the complex formation with co-catalysts and electron donors, and a certain proportion of these metal atoms included in the complex is active for polymerization [3]. There are many various reasons for the real or visible deactivation of the active centers. The rate constants of various decontamination processes can be concentrated in a single coefficient $k_{d}$, while the decrease in the concentration of active centers C* can be assessed using the ratio $C^{*}=C_{0}^{*}e^{-k_{d}t}$, even if it is caused by other processes [4].

In [5], the polymerization kinetics was studied for various temperatures, pre-polymerization methods, and hydrogen and monomer concentrations for polymerization in a liquid propylene medium in industrially significant conditions using a TiCl_4_/phthalate/MgCl_2_ + AlEt_3_/DCPDMS catalyst (DCPDMS-dicyclopentyl-dimethoxy-silane). It is found that the activation energies observed for the polymerization experiments depend on the method used to calculate the monomer concentration on active centers, since such methods suggest their own temperature dependences. In all the cases considered in [5], the observed activation energy was relatively high, and the polymerization rate was not limited by the monomer transfer into the particles (activation energy for the initial reaction rate $E_{p0}$ = 315.93 cal/mol and the deactivation constant E_d_ = 179.75 cal/mol for the main polymerization at 0.21 moles of hydrogen and the monomer concentration calculated from the liquid density, after fixed pre-polymerization, i.e., $E_{p0}/R\approx$ 9073.4, $E_{d}/R\approx$ 5162, *R* being the molar gas constant). At high initial reaction rates, a pre-polymerization is necessary to prevent a thermal acceleration (a sharp increase in the temperature) for the largest catalyst particles. It was shown [5] that the polymerization temperature significantly affects the morphology of polymer particles. At low temperatures, regularly shaped particles were obtained, characterized by a high density and low porosity. As the temperature increased, the morphology gradually turned into a more open structure with uneven surfaces and a poor reproduction of the shapes of the catalyst particles.

The polymerization temperature effect can be explained in different ways; an increase in the temperature causes an increase in the reaction rate in the catalyst/polymer particles, while the following is possible: (1) overheating the particles and forming gas bubbles and, therefore, more open surface structures; (2) the uncontrolled fragmentation of the catalyst carrier (substrate); and (3) changes in the physical properties of the polymer, expanding by monomer, fluxing, and changing the particle shapes.

Detailed knowledge of the olefin polymerization kinetics using the Ziegler–Nattat-type TMC is crucial for modeling the industrial processes of polyolefin synthesis.

The literature review has shown that the process rate is determined by the kinetics of propylene polymerization on the TMC during the PP synthesis, which ensures intensive mixing in the polymerization equipment and the efficient reaction of the heat removal [6]. Single-center kinetic models approximate the experimental values of the PP yield (Yp,e) and its number average molecular weight (Mn,e) properly, but the mass average molecular weight (Mw,e) and the polydispersity index (Kwn,e) cannot be approximated. Two- and three-center kinetic models approximate the experimental values of Yp,e, Mn,e, and Mw,e properly, but they poorly approximate the experimental PP MWD. The deconvolution of the experimental PP MWD showed the presence of 4–5 types of active centers that produced the polymer chains of different lengths at different rates [7]. Polycentric kinetic models containing many kinetic parameters can describe experimental data (Yp,e, Mn,e, Mw,e, and Kwn,e) even with the values of the kinetic parameters, which have no physical meaning.

There are the following methods for constructing a polycentric kinetic model and identifying its parameters:Using simplified kinetic schemes, including the minimum required set of the elementary stages of the polymerization process.Writing the material balances of reactants using simplifying assumptions (resistances to heat and mass transfer inside the catalyst/polymer particles are not taken into account, and the monomer concentration at the AC is assumed to be equal to the concentration in the core of the liquid phase, etc.).Assuming that the PP MWD is a superposition of the Flory distributions of polymer chains produced on each selected center type.Identifying the kinetic parameters in stages.

This paper is aimed at developing a polycentric model of the PP polymerization liquid-phase kinetics to the TiCl_4_/DBP/MgCl_2_ + TEA/CHMDMS (DBP—dibutyl phthalate, TEA—triethylaluminium, CHMDMS—cyclohexyl-methyl-dimethoxy-silane) catalyst system to predict the polymerization rate, the polymer yield, and the average molecular weight of polymer chains where the polymerization temperature ranges from 40 to 90 °C, and to present an algorithm for a phased evaluation of the kinetic parameters based on the experimental data.

## 2. Kinetic Model of TMC-Based Propylene Polymerization

When modeling the TMC-based PP polymerization, the following assumptions are made: (1) the reaction is carried out under isothermal and isobaric conditions; (2) during polymerization in a liquid propylene medium, the monomer concentration is constant; (3) with a sufficiently high reaction mixture recirculation degree, the reactor can be considered as an ideal mixing reactor; (4) the resistance to the mass and heat transfer, as well as a diffusion of reagents, are not taken into account due to good mixing; (5) the polymerization process is limited by the process kinetics; (6) the monomer concentration on the AC is assumed to be equal to the monomer volume concentration; (7) there is no polar impurity in the reaction medium; and (8) the chain transfer reactions form the same center type that was originally formed in activating the catalyst by a co-catalyst.

There are many research studies about the kinetics of a propylene polymerization on Ziegler–Natta catalysts. It was shown in [8,9,10] that the polymerization rate under isothermal conditions can be described as the first-order process in the monomer concentration, and the catalyst deactivation rate as the first-order process in active center concentration. The kinetics of the TMC-based propylene polymerization in its simplest form can be described using a set of elementary reaction steps for each site type. To construct a mathematical model of the liquid-phase polymerization kinetics of propylene on the heterogeneous catalytic system TiCl_4_/DBP/MgCl_2_ + TEA/CHMDMS, reflecting the temperature dependence of the kinetic parameters, a kinetic scheme was used [11] (Table 1):

Where j is the center type; $C_{j}$ is a potential active center; A, M, and H_2_ are the co-catalyst, monomer, and hydrogen, respectively; $P_{j,0}$ is the vacant monomer center; $P_{j,r}$ is the living polymer chain of the length r; $D_{j,r}$ is the dead polymer chain of the length r (r is the number of monomer units in the chain, chain length, and degree of polymerization), synthesized on the AC of the j^th^-type; $k_{a,j},{\;k}_{i,j},k_{p,j},k_{d,j},k_{m,j},k_{H,j}$ are the rate constants of the activation, initiation, growth, deactivation, and chain transfer to the monomer and hydrogen for the j^th^-type AC; and $N_{s}$ is the number of AC types. Due to the identification complexity, the initiation rate is usually considered to be equal to the growth speed of the polymer chain $k_{i,j}=k_{p,j}$.

The material balances of the components included in the kinetic scheme are written as:(1)$$\frac{\partial M\left(t,T\right)}{\partial t}=-M\left(t,T\right)\sum_{j=1}^{N_{s}}k_{\mathrm{pj}}\left(T\right)\sum_{r=0}^{\infty }P_{j,r}\left(t,T\right),\;M\left(0,T\right)=M_{0}\left(T\right),$$
(2)$$\frac{\partial H\left(t,T\right)}{\partial t}=-H\left(t,T\right)\sum_{j=1}^{N_{s}}k_{\mathrm{Hj}}\left(T\right)\sum_{r=0}^{\infty }P_{j,r}\left(t,T\right),\;H\left(0,T\right)=H_{0}\left(T\right),$$
(3)$$\frac{\partial A\left(t,T\right)}{\partial t}=-A\left(t,T\right)\sum_{j=1}^{N_{s}}k_{\mathrm{aj}}\left(T\right)C_{j}\left(t,T\right),\;A\left(0,T\right)=A_{0}\left(T\right),$$
(4)$$\frac{\partial C_{j}\left(t,T\right)}{\partial t}=-k_{\mathrm{aj}}\left(T\right)A\left(t,T\right)C_{j}\left(t,T\right),\;C_{j}\left(0,T\right)=C_{j0}\left(T\right),$$
(5)$$\frac{\partial P_{j,0}\left(t,T\right)}{\partial t}=-k_{\mathrm{aj}}\left(T\right)A\left(t,T\right)C_{j}\left(t,T\right)-k_{\mathrm{pj}}\left(T\right)M\left(t,T\right)P_{j,0}\left(t,T\right)-k_{\mathrm{dj}}\left(T\right)P_{j,0}\left(t,T\right)+\;k_{\mathrm{Hj}}\left(T\right)H\left(t,T\right)\overset{\infty }{\underset{r=1}{\sum }}P_{j,r}\left(t,T\right),$$
(6)$$\frac{\partial P_{j,1}\left(t,T\right)}{\partial t}=-k_{\mathrm{pj}}\left(T\right)M\left(t,T\right)(P_{j,0}\left(t,T\right)-P_{j,1}\left(t,T\right))+k_{\mathrm{mj}}\left(T\right)M\left(t,T\right)\overset{\infty }{\underset{r=1}{\sum }}P_{j,r}\left(t,T\right)-\;\left(k_{\mathrm{Hj}}\left(T\right)H\left(t,T\right)+k_{\mathrm{mj}}\left(T\right)M\left(t,T\right)+k_{\mathrm{dj}}\left(T\right)\right)P_{j,1}\left(t,T\right),$$
(7)$$\frac{\partial P_{j,r}\left(t,T\right)}{\partial t}=k_{\mathrm{pj}}\left(T\right)M\left(t,T\right)(P_{j,r-1}\left(t,T\right)-P_{j,r}\left(t,T\right))-\;\left(k_{\mathrm{Hj}}\left(T\right)H\left(t,T\right)+k_{\mathrm{mj}}\left(T\right)M\left(t,T\right)+k_{\mathrm{dj}}\left(T\right)\right)P_{j,r}\left(t,T\right),$$
(8)$$\frac{\partial D_{j,r}\left(t,T\right)}{\partial t}=\left(k_{\mathrm{Hj}}\left(T\right)H\left(t,T\right)+k_{\mathrm{mj}}\left(T\right)M\left(t,T\right)+k_{\mathrm{dj}}\left(T\right)\right)P_{j,r},\left(t,T\right),\;\;j=\bar{1,N_{s}},\;r=1,2,\ldots$$

In Equations (1)–(8) A, M, and H are the concentrations of the co-catalyst, monomer, and hydrogen in the liquid phase and $P_{j,r}$ and $D_{j,r}$ are the concentrations of living and dead polymer chains, respectively.

The method of moments was used to calculate the polymerization rate and the molecular weight characteristics of PP. Basically, it is sufficient to confine to the knowledge of the statistical moments of the first few orders, which provide reasonable information for many practical purposes about the distribution nature. Therefore, the polymerization rate and the number average molecular weight are determined by zero- and first-order moments and, to determine the polydispersity index, it is essential to know the second-order moments. Balance equations were obtained for the distribution moments of living $Y_{j}^{n}$ and dead $X_{j}^{n}$ polymer chains along the length, as follows:(9)$$\frac{\partial Y_{j}^{\left(0\right)}\left(t,T\right)}{\partial t}=k_{\mathrm{aj}}\left(T\right)A\left(t,T\right)C_{j}\left(t,T\right)-k_{\mathrm{dj}}\left(T\right)Y_{j}^{\left(0\right)}\left(t,T\right),$$
(10)$$\frac{\partial Y_{j}^{\left(1\right)}\left(t,T\right)}{\partial t}=k_{\mathrm{pj}}\left(T\right)M\left(t,T\right)Y_{j}^{\left(0\right)}\left(t,T\right)+k_{\mathrm{mj}}\left(T\right)M\left(Y_{j}^{\left(0\right)}\left(t,T\right)-P_{j,0}\left(t,T\right)\right)\;-(k_{\mathrm{Hj}}\left(T\right)H\left(t,T\right)+{\;k}_{\mathrm{mj}}\left(T\right)M\left(t,T\right)+k_{\mathrm{dj}}\left(T\right)Y_{j}^{\left(1\right)}\left(t,T\right),$$
(11)$$\frac{\partial Y_{j}^{\left(2\right)}\left(t,T\right)}{\partial t}=k_{\mathrm{pj}}\left(T\right)M\left(t,T\right)\left(2Y_{j}^{\left(1\right)}\left(t,T\right)+Y_{j}^{\left(0\right)}\left(t,T\right)\right)+\;k_{\mathrm{mj}}\left(T\right)M\left(t,T\right)\left(Y_{j}^{\left(0\right)}\left(t,T\right)-P_{j,0}\left(t,T\right)\right)$$
(12)$$-\left(k_{\mathrm{Hj}}\left(T\right)H\left(t,T\right)+k_{\mathrm{mj}}\left(T\right)M\left(t,T\right)+k_{\mathrm{dj}}\left(T\right)\right)Y_{j}^{\left(2\right)}\left(t,T\right),\;\frac{\partial X_{j}^{\left(n\right)}\left(t,T\right)}{\partial t}=\left(k_{\mathrm{Hj}}\left(T\right)H\left(t,T\right)+k_{\mathrm{mj}}\left(T\right)M\left(t,T\right)+k_{\mathrm{dj}}\left(T\right)\right)Y_{j}^{\left(n\right)}\left(t,T\right),$$
(13)$$Y_{j}^{\left(n\right)}\left(0,T\right)=0,X_{j}^{\left(n\right)}\left(0,T\right)=0,\;n=0,1,2,\;j=\bar{1,N_{s}}$$
where $Y_{j}^{\left(n\right)}\left(t,T\right)=\sum_{r=1}^{\infty }r^{n}P_{j,r}\left(t,T\right)$ is the nth moment of distribution along the length (polymerization degree) of living polymer chains associated with the j-type active center, n = 1,2,...; $Y_{j}^{\left(0\right)}\left(t,T\right)=\sum_{r=0}^{\infty }P_{j,r}\left(t,T\right)$, since $k_{i,j}=k_{p,j}$ and $X_{j}^{\left(n\right)}\left(t,T\right)=\sum_{r=1}^{\infty }r^{n}D_{j,r}\left(t,T\right)$ is the nth moment of distribution along the length of the dead polymer chains synthesized on the j-type active centers.

By a model that includes Equations (1)–(5) and (9)–(13), it was calculated:

polymerization rate:(14)$$R_{p}\left(t,T\right)=V_{R}\left(T\right)M\left(t,T\right)\sum_{j=1}^{N_{s}}k_{\mathrm{pj}}\left(T\right)Y_{j}^{\left(0\right)}\left(t,T\right),$$
yield of the fraction of the polymer synthesized at each type of centers
(15)$$Y_{p,j}\left(t,T\right)={\mathrm{mw}}_{m}V_{R}\left(T\right)\left(X_{j}^{\left(1\right)}\left(t,T\right)+Y_{j}^{\left(1\right)}\left(t,T\right)\right),$$
mass fraction of the polymer fraction
(16)$$p_{j}\left(t,T\right)=\frac{Y_{p,j}\left(t,T\right)}{\sum_{i=1}^{N_{s}}Y_{p,j}\left(t,T\right)},\;j=\bar{1,N_{s}}$$
average molecular weights and polydispersity index of the polymer
(17)$$M_{n}\left(t,T\right)={\mathrm{mw}}_{m}\frac{\sum_{j=1}^{N_{s}}\left(X_{j}^{\left(1\right)}\left(t,T\right)+Y_{j}^{\left(1\right)}\left(t,T\right)\right)}{\sum_{j=1}^{N_{s}}\left(X_{j}^{\left(0\right)}\left(t,T\right)+Y_{j}^{\left(0\right)}\left(t,T\right)\right)},$$
(18)$$M_{w}\left(t,T\right)={\mathrm{mw}}_{m}\frac{\sum_{j=1}^{N_{s}}\left(X_{j}^{\left(2\right)}\left(t,T\right)+Y_{j}^{\left(2\right)}\left(t,T\right)\right)}{\sum_{j=1}^{N_{s}}\left(X_{j}^{\left(1\right)}\left(t,T\right)+Y_{j}^{\left(1\right)}\left(t,T\right)\right)},$$
(19)$$K_{\mathrm{wn}}\left(t,T\right)=\frac{M_{w}\left(t,T\right)}{M_{n}\left(t,T\right)}$$
$V_{R}\left(T\right)$ is the reaction mixture volume and $mw_{m}$ is the molecular weight of propylene.

For a step-by-step identification of the temperature dependence of the kinetic parameters, the following transformations are performed. Since the co-catalyst is present in excess [1], the product $k_{aj}A$ = $K_{Aj}$ was considered constant, and it results from the relations (4):(20)$$C_{j}\left(t,T\right)=C_{j,0}\left(T\right)e^{-K_{\mathrm{Aj}}\left(T\right)t},\;\;j=\bar{1,N_{s}}$$

The integration of (9) under the initial condition led to the following expression:(21)$$Y_{j}^{\left(0\right)}\left(t,T\right)=\frac{C_{j,0}\left(T\right)}{1-\frac{k_{\mathrm{dj}}\left(T\right)}{K_{\mathrm{Aj}}\left(T\right)}}\left(e^{-k_{\mathrm{dj}}\left(T\right)t}-e^{-K_{\mathrm{Aj}}\left(T\right)t}\right)$$

Substituting (21) into the Formula of (14), the expression for calculating the monomer polymerization rate was transformed as follows:(22)$$R_{p}\left(t,T\right)=V_{R}\left(T\right)M\left(t,T\right)\sum_{j=1}^{N_{s}}[\frac{k_{\mathrm{pj}}\left(T\right)C_{j,0}\left(T\right)}{1-\frac{k_{\mathrm{dj}}\left(T\right)}{K_{\mathrm{Aj}}\left(T\right)}}\times \left(e^{-k_{\mathrm{dj}}\left(T\right)t}-e^{-K_{\mathrm{Aj}}\left(T\right)t}\right)],\;$$

The total polymer yield can be calculated by integrating the Expression (22) over the polymerization time, $t_{p}$:(23)$$Y_{p}\left(t,T\right)={\mathrm{mw}}_{m}\int_{0}^{t_{p}}R_{p}\left(t,T\right)\mathrm{dt},\;$$

During polymerization in a liquid monomer, its concentration was considered to be constant $M\left(t,T\right)=M_{0}\left(T\right)$.

## 3. Computing the Temperature Dependence of Kinetic Parameters: Algorithm and Results

Stage 1: Determining the activation energies for a single-center kinetic model.

For a single-center model of the propylene polymerization process at 70°, the values of $k_{p}^{\left(70\right)}$ = 6120 L/(mol min), $K_{A}^{\left(70\right)}$ = 3 min^−1^, $k_{d}^{\left(70\right)}$ = 0.018 min^−1^, $k_{m}^{\left(70\right)}$ = 2.4, and $k_{h}^{\left(70\right)}$ = 280 L/(mol min) [11] are found, which are selected as the reference, and their temperature dependence is represented as follows:(24)$$k_{p}\left(T\right)=k_{p}^{\left(70\right)}e^{-\frac{E_{p}}{R}\left(\frac{1}{T}-\frac{1}{343}\right)},\;K_{A}\left(T\right)=K_{A}^{\left(70\right)}e^{-\frac{E_{a}}{R}\left(\frac{1}{T}-\frac{1}{343}\right)},\;k_{d}\left(T\right)=k_{d}^{\left(70\right)}e^{-\frac{E_{d}}{R}\left(\frac{1}{T}-\frac{1}{343}\right)},\;k_{m}\left(T\right)=k_{m}^{\left(70\right)}e^{-\frac{E_{m}}{R}\left(\frac{1}{T}-\frac{1}{343}\right)}\;k_{H}\left(T\right)=k_{H}^{\left(70\right)}e^{-\frac{E_{h}}{R}\left(\frac{1}{T}-\frac{1}{343}\right)}$$

Using the experimental data from [1] given in Table 1, we constructed the dependence $ln\left(Y_{p}/\rho_{m}\right)$ on 1/T and identified the effective activation energy of the polymerization process $E_{ef}^{ex}$ = 10037.8 cal/mol. It was considered that, in the case of any changes in the polymerization temperature, the concentration (density $\rho_{m}$) of the monomer also changes, which affects the polymerization rate and the PP yield.

Using $E_{ef}^{ex}$ as an initial approximation ($E_{n}=E_{ef}^{ex}$) and using the NMinimize subroutine of the Mathematica package, we found the minimum of the function:(25)$$\underset{E_{p},{\;E}_{a},{\;E}_{d}}{\min}\sum_{i=1}^{N_{T}}{\left(Y_{p}\left(t_{p},T_{i},E_{p},{\;E}_{a},{\;E}_{d}\right)-Y_{p,e}\left(t_{p},T_{i}\right)\right)}^{2},\;$$

In which the polymer outputs are represented as:(26)$$\begin{array}{cc} Y_{p}\left(t_{p},T_{i},E_{p},E_{a},E_{d}\right)\\ \; & =\frac{\rho_{m}\left(T_{i}\right)m_{\mathrm{cat}}f_{\mathrm{Ti}}k_{p}\left(T_{i}\right)}{{\mathrm{mw}}_{\mathrm{Ti}}\left(1-k_{d}\left(T_{i}\right)/K_{A}\left(T_{i}\right)\right)}\\ \; & \times \left[\frac{1}{k_{d}\left(T_{i}\right)}\left(1-e^{-k_{d}\left(T_{i}\right)t_{p}}\right)-\frac{1}{\;K_{A}\left(T_{i}\right)}\left(1-e^{-K_{A\left(T_{i}\right)}t_{p}}\right)\right] \end{array}$$

The restrictions are written as $k_{1}E_{n}<E_{l}<k_{2}E_{n}$ (l=p,a,d, coefficients k_1_ = 0.1, k_2_ = 5, and N_T_ = 6), and the polymerization duration t_p_ = 120 min. In Formula (26), $m_{cat}$ is the catalyst mass, $f_{Ti}$ is the mass fraction of titanium in the catalyst, and $mw_{Ti}$ is the molecular weight of titanium ($Y_{p,e}\left(t_{p},T_{i}\right)={\tilde{Y}}_{p,e}\left(t_{p},T_{i}\right)m_{cat}$). As a result, the activation energies of the chain growth reactions E_p_ = 20,564 cal/mol, activation E_a_ = 21,815 cal/mol, and deactivation E_d_ = 20,331 cal/mol AC were found for a single-center kinetic model.

In [2], it was found experimentally that the effective activation energy of the olefin (ethylene) polymerization process E_ef_ differs from the activation energy of the growth reaction, E_p_. The difference between the values of E_p_ and E_ef_ is caused by a change in the AC concentration having the temperature that is numerically expressed by the temperature coefficient, $E_{c^{*}}$.

In the single-center model, the formula is valid for the polymer yield Y_p_:(27)$$Y_{p}\left(t_{p},T_{i}\right)=V_{R}\left(T_{i}\right)\int_{0}^{t_{p}}k_{p}\left(T_{i}\right)M\left(T_{i}\right)Y^{0}\left(t,T_{i}\right)\mathrm{dt}\approx V_{R}\left(T_{i}\right)k_{p}\left(T_{i}\right)M\left(T_{i}\right){\bar{Y}}^{0}\left(T_{i}\right)t_{p},$$
where ${\bar{Y}}^{\left(0\right)}\left(T_{i}\right)$ is the average AC concentration of the AC during polymerization t_p_; Formula (27) is written considering that during polymerization in a liquid monomer, the values of M and k_p_ do not depend on the process duration t_p_.

Since the AC concentration at time t is calculated by the Formula (21) at $N_{s}$ = 1, the average AC concentration value during polymerization t_p_ was calculated by the formula as follows:(28)$${\bar{Y}}^{\left(0\right)}\left(T_{i}\right)=\int_{0}^{t_{p}}Y^{\left(0\right)}\left(t,T_{i}\right)\mathrm{dt}/t_{p}=\frac{C_{0}\left(T_{i}\right)}{(1-k_{d}\left(T_{i}\right)/K_{A}\left(T_{i}\right))t_{p}}\left[\frac{1}{k_{d}\left(T_{i}\right)}(1-e^{-k_{d}\left(T_{i}\right)t}\right)-\frac{1}{K_{A}\left(T_{i}\right)}(1-e^{-K_{A}\left(T_{i}\right)t})].$$
when integrating, it was considered that K_A_ and k_d_ do not depend on t_p_.

It was found in [2] that the AC concentration in the suspension process of olefin polymerization depends considerably on the process temperature and the following formula is applicable:(29)$${\bar{Y}}^{\left(0\right)}\left(T_{i}\right)={\bar{Y}}_{0}^{\left(0\right)}e^{-\frac{E_{c^{*}}}{{\mathrm{RT}}_{i}}}$$
for the constant of the speed of the growth of chains, it can also be written:(30)$$k_{p}\left(T_{i}\right)=k_{p,0}e^{-\frac{E_{p}}{{\mathrm{RT}}_{i}}},\;\mathrm{where}{\;k}_{p,0}=k_{p}^{\left(70\right)}e^{\frac{E_{p}}{R\left(273+70\right)}},$$
thus, from the Formulas (27), (29), and (30), it should be:(31)$$Y_{p}\left(t_{p},T_{i}\right)\approx V_{R}\left(T_{i}\right)M\left(T_{i}\right)k_{p,0}{\bar{Y}}_{0}^{\left(0\right)}e^{-\frac{\left(E_{p}+E_{c^{*}}\right)}{{\mathrm{RT}}_{i}}}t_{p}.$$

It can be seen from Formula (31) that within the framework of the single-center model which is presented, the effective activation energy of the polymerization process E_ef_ is equal to
(32)$$E_{\mathrm{ef}}=E_{p}+E_{c^{*}}$$

The approximation of the dependence $ln{\bar{Y}}^{\left(0\right)}$ on 1/T using Formula (28) and the Fit subroutine of the Mathematica package allowed us to find the value of the temperature coefficient $E_{c^{*}}$ = −12211.9 cal/mol. Thus, the calculated value of the effective activation energy of the polymerization process is E_ef_ = E_p_ + $E_{c^{*\;}}$ = 20564–12211.9 = 8352.1 cal/mol. Comparing the effective activation energies of the polymerization process experimental $E_{ef}^{ex}$ = 10037.8 cal/mol and the calculated $E_{ef}$ one, we obtained a relative error that is equal $\Delta E_{ef}=\left|E_{ef}^{ex}-E_{ef}\right|/E_{ef}^{ex}$ to = 0.168.

It should be noted that according to the experimental data of [2], with an increase in the ethylene polymerization temperature, the AC concentration increased for the reaction duration of t_p_ = 5–15 min, there is no information for t_p_ = 60–120 min. Calculations based on a single-center model of propylene polymerization on the TMC studied show a change in the nature of the AC concentration dependence on the reaction temperature (in the range of 40–90 °C) with an increase in the polymerization duration. At t_p_ = 1 min., ${\bar{Y}}^{\left(0\right)}$ increases with the temperature growing from 40 to 80 °C, and ${\bar{Y}}^{\left(0\right)}$ decreases within the range of 80–90 °C; at t_p_ = 5 min. ${\bar{Y}}^{\left(0\right)}$ increases with a change in the temperature from 40 to 70 °C and further decreases.

Similarly, using $E_{ef}^{ex}$ as an initial approximation ($E_{n}=E_{ef}^{ex}$) and the NMinimize subroutine of the Mathematica package, the minimum of the function is found:(33)$$\underset{{\;E}_{m},{\;E}_{h}}{\min}\overset{N_{T}}{\underset{i=1}{\sum }}{\left(M_{n}\left(t_{p},T_{i},E_{m},{\;E}_{h},\right)-M_{n,e}\left(t_{p},T_{i}\right)\right)}^{2}$$

The restrictions are written as $k_{1}E_{n}<E_{l}<k_{2}E_{n}$ (l=m,h, coefficients k_1_ = 0.1, k_2_ = 5, and N_T_ = 6), and the duration of polymerization t_p_ = 120 min. As a result, the activation energies of the chain transfer reactions to the monomer E_m_ and hydrogen E_h_ were determined for a single-center kinetic model ($E_{m}$ = 12,319.6 $E_{h}$ = 35753.8 cal/mol).

It is interesting to compare the values of the activation energies obtained in this work for the kinetic coefficients of a single-center model of propylene polymerization on the TMC under consideration ($E_{p}/R\approx$ 10,386, $E_{d}/R\approx$ 10,268, and $E_{ef}^{ex}/R\approx$ 5070) with the published data. In [12,13], kinetic coefficients $k_{a}$, $k_{p}$, $k_{H}$, $k_{m}$, $k_{d}$, were used to calculate the liquid-phase polymerization of propylene, the temperature dependences of which were determined by the Arrhenius law with the same activation energy E = 12,000 cal/mol, i.e., E/R ≈ 6030, which correlates with $E_{ef}^{ex}/R\approx$ 5070. It should be noted that the value of the effective activation energy of the polymerization process depends on the interval of the change in the polymerization temperature used to determine it. In [14], the liquid-phase polymerization of propylene was studied in a loop reactor on TiCl_4_/MgCl_2_ + PEEB + AlR_3_ catalyst, (PEEB—paraethoxyethyl benzoate), at a temperature of 70 °C using a mathematical model, all the kinetic coefficients of which had an activation energy E = 209.5 cal/mol (E/R ≈ 6017). The same activation energy was used for the temperature dependence of all the kinetic coefficients of the model that is used for the analysis of the polyolefin production process using the Spheripol technology in [7]. In [6], for the temperature dependence of the kinetic coefficients $k_{i}$, $k_{p}$, $k_{H}$, $k_{d}$, a single activation energy E = 217,46 cal/mol (E/R ≈ 6245) was used, as well as in the articles [8,9]. Yang et al. [6] concluded that the ideal mixing model (CSTR) could be used to describe the flow structure in a loop reactor of propylene polymerization at a recirculation coefficient above 50. The reactor model presented in [10] includes kinetic coefficients with activation energies of $E_{a}$ = $E_{d}$ = 12,000, $E_{p}$ = 10,000, and $E_{m}$ = $E_{h}$ = 14,000 cal/mol. In [15], the liquid-phase polymerization of propylene with a highly active catalyst TiCl_4_/phthalate/MgCl_2_ + TEA/silane was studied in a filled intermittent reactor at a pressure of 43 bar, at the temperatures of 60–80 °C, using a quasi-one-center model based on the mechanism of dormant centers. The calculated effective activation energies were 245.53, 266.48, and 237.57 cal/mol for 0.0, 150, and 1000 mg of hydrogen, respectively, in the temperature range of 60–80 °C ($E_{ef}/R\approx$ 7052). The authors [15] emphasize that the chain growth rate constant $k_{p}$ is averaged due to the heterogeneity of the AC. The true amount of AC and the sorption of the monomer were not considered, although they affect the value $k_{p}$ ($E_{p}$ = 281.65 cal/mol, $E_{p}/R\approx$ 8089). In [3], for the liquid-phase polymerization of propylene using the TiCl_4_/phthalate/MgCl_2_ + TEA/silane catalyst, the apparent activation energy of the growth reaction $E_{p}$ = 272.77 cal/mol and the deactivation reaction $E_{d}$ = 103.49 cal/mol were assessed according to the Arrhenius graph for the initial polymerization rate.

Thus, the value of $\frac{E_{p}}{R}\approx$ of 10,386 obtained in this work is the closest to the value of $\frac{E_{p0}}{R}\approx$ of 9073.4 from the article [5]; when $E_{p0},$ it was also determined for the concentration of the monomer on the AC calculated from the liquid propylene density.

Stage 2: Deconvolution of experimental PP MWD

To identify the temperature dependence of the parameters of the polycentric model of polymerization kinetics, the experimental MWD of PP samples synthesized at temperatures T = 40–90 °C [1] are presented as the superpositions of the Flory distributions of polymer chains produced on each AC type [16].

Figure 1 shows the MWD deconvolutions of the PP samples obtained by polymerization at the temperatures studied (T = 40–90 °C). The mass fractions $p_{j}$ and the average calculated molecular weights $M_{nj}$ of the polymer fractions that had been synthesized on the selected AC types were found by the methods proposed in [16]. The selected five types of AC are numbered in ascending order, $M_{nj}$, of the polymer fractions. The positions of the Flory distribution peaks of each fraction do not change much, but their relative proportions do. Some of the changes in the average molecular weights with an increasing polymerization time are caused by the fragmentation and growth of catalyst/polymer particles. Relative proportions of the resulting polymer fractions change, because all types of centers have different polymerization kinetics. The balance between the activation, growth, and deactivation rates that are quantified by kA, kp, and kd for each center type determines the change in the mass fractions of the polymer fractions depending on the polymerization time, and it also determines the time evolution of the PP MWD in these polymerizations.

Figure 1 shows that when the polymerization temperature changes, the mass fractions of the polymer fractions formed a change as well, since the kinetic parameters ($K_{Aj},\;k_{pj},k_{dj}$, $j=\bar{1,N_{s}}$) of different AC types have different temperature dependences. When presenting the experimental MWD of the PP samples as a superposition of the Flory distributions of polymer chain fractions, there are usually 3–5 types of ACs producing these fractions, the largest number of which corresponds with the wider ranges of changes in the polymerization temperature and hydrogen concentration.

Stage 3: Determination of activation energies for the polycenteric model of kinetics

Using $E_{a},E_{p}$, $\;E_{d}$ as initial approximations $E_{an}$, $\;E_{pn}$, $\;E_{dn}$, and the NMinimize subroutine of the Mathematica package, the minimum of the function was found:(34)$$\underset{\begin{array}{c} E_{\mathrm{pj}},E_{\mathrm{aj}},{\;E}_{\mathrm{dj}}\\ j=\bar{1,\mathrm{Ns}} \end{array}}{\min}\sum_{i=1}^{N_{T}}\sum_{j=1}^{\mathrm{Ns}}{\left(\begin{array}{c} Y_{\mathrm{pj}}\left(t_{p},{\;T}_{i},E_{\mathrm{pj}},E_{\mathrm{aj}},{\;E}_{\mathrm{dj}}\right)-\\ Y_{\mathrm{pj},c}\left(t_{p},T_{i}\right) \end{array}\right)}^{2},$$
in which the yields of polymer fractions were calculated by the formula:(35)$$Y_{\mathrm{pj}}\left(t_{p},T_{i},{\;E}_{\mathrm{pj}},{\;E}_{\mathrm{aj}},{\;E}_{\mathrm{dj}}\right)=\frac{\rho_{m}\left[T_{i}\right]m_{\mathrm{cat}}f_{\mathrm{Ti}}k_{\mathrm{pj}}\left(T_{i}\right)}{{\mathrm{mw}}_{\mathrm{Ti}}\left(1-k_{\mathrm{dj}}\left(T_{i}\right)/K_{\mathrm{Aj}}\left(T_{i}\right)\right)}\times \left[\frac{1}{k_{\mathrm{dj}}\left(T_{i}\right)}\left(1-e^{-k_{\mathrm{dj}}\left(T_{i}\right)t_{p}}\right)-\;\frac{1}{K_{\mathrm{Aj}}\left(T_{i}\right)}\left(1-e^{-K_{\mathrm{Aj}}\left(T_{i}\right)t_{p}}\right)\right],$$
the restrictions are written in the form $k_{1}E_{ln}<E_{lj}<k_{2}E_{ln}$ (l=p,a,d, coefficients k_1_ = 0.1, k_2_ = 5, N_T_ = 6), and the duration of polymerization t_p_ = 120 min. As a result, the activation energies of the chain growth reactions $E_{pj}$, the activation $E_{aj},$ and the deactivation of active centers were determined $E_{dj}$ for a polycenter kinetic model (Table 4).

The values of the yields of the PP fractions synthesized at the studied temperatures were determined by the formulas:(36)$$Y_{\mathrm{pj},c}\left(t_{p},T_{i}\right)={\tilde{Y}}_{p,e}\left(t_{p},T_{i}\right)m_{\mathrm{cat}}p_{j}\left(T_{i}\right),\;j=\bar{1,N_{s}},\;i=\bar{1,N_{T}},$$
using the corresponding values ${\tilde{Y}}_{p,e}\left(t_{p},T_{i}\right)$ and $p_{j}\left(T_{i}\right)$ from Table 1 and Table 2 ($t_{p}$ = 120 min).

The preexponents in the Arrhenius dependence were calculated by the formula:(37)$$k_{\mathrm{lj}0}=k_{\mathrm{lj}}^{\left(70\right)}e^{\frac{E_{\mathrm{lj}}}{R\left(273+70\right)}},\;l=a,p,d,h,m$$

Figure 2 shows that the AC concentrations $Y^{\left(0\right)}$ at a temperature above 80 °C is significantly reduced during the polymerization, which leads to a slowdown in the polymer formation rate (Figure 3).

Using $E_{m}$ and $E_{h}$ as an initial approximation ($E_{mn}$ = $E_{m}$, $E_{hn}$ = $E_{h}$) and the NMinimize subroutine of the Mathematica package, the minimum of the function was found:(38)$$\underset{\begin{array}{c} E_{\mathrm{mj}},{\;E}_{\mathrm{hj}}\\ j=\bar{1,\mathrm{Ns}} \end{array}}{\min}\sum_{i=1}^{N_{T}}\sum_{j=1}^{\mathrm{Ns}}{\left(\begin{array}{c} M_{\mathrm{nj}}\left(t_{p},T_{i},{\;E}_{\mathrm{mj}},E_{\mathrm{hj}}\right)-\\ M_{\mathrm{nj},c}\left(t_{p},T_{i}\right) \end{array}\right)}^{2},\;$$

Restrictions are written as $k_{1}E_{ln}<E_{lj}<k_{2}E_{ln}$ (l=m,h, coefficients k_1_ = 0.1, k_2_ = 5, and N_T_ = 6), and the polymerization duration t_p_ = 120 min. As a result, the activation energies of the chain transfer reactions to monomer $E_{mj}$ and hydrogen were determined $E_{hj}$ for a polycentric kinetic model. The values of $M_{nj,c}\left(t_{p},T_{i}\right)$ are shown in Table 2 (t_p_ = 120 min.).

Dependencies $Y_{j}^{\left(n\right)}\left(t,T\right)$, $X_{j}^{\left(n\right)}\left(t,T\right)$, (n=0,1,2), are calculated according to the polycenteric model developed and can also be represented as 3D graphs (Figure 4, Figure 5 and Figure 6).

Figure 7 and Figure 8 show the efficacy of the presented polycentric model of the liquid-phase propylene polymerization kinetics on the TMC considered and of the algorithm for identifying the temperature dependence of the kinetic coefficients.

For polycentric models of propylene polymerization kinetics at TMC, not so many sets of activation energies for the Arrhenius dependences of kinetic coefficients have been published. In [17,18,19,20,21], the kinetic constants of a propylene polymerization in a liquid monomer were quantified using the TiCl_4_/EB/MgCl_2_ + DEAC + TEA/PEEB catalyst (EB is ethyl benzoate, PEEB is para-ethoxy-ethyl benzoate, DEAC-diethyl-aluminum chloride; titanium content ~ 3% by weight; and 60 ≤ T ≤ 70 °C). The authors focused the dependence of the polypropylene yield on the temperature in the growth constant and obtained the values for the single-center model: kp = exp [A-E_A_/T], A = 2.860, and E_A_ = 1977. To approximate the experimental MWD of the polymer samples, two center types were required, for which the following was found: A_1_ = 0.677, E_A1_ = 1201.3; and A_2_ = 9.78, E_A2_ = 4365.1. The authors of [18] investigated the semi-batch suspension polymerization of propylene using the catalyst system of TiCl_4_/DiBPH/MgCl_2_ + TEA/DiMECHS which was researched (DiMECHS is dimethoxy-methyl-cyclohexyl silane, DiBPH is dibutyl phthalate; the titanium content of 2.7 wt.%; Al/Si = 20, Al/Ti = 340–500; the solvent is *n*-heptane; T= 60–80 °C, P = 7 bar; and the volume of hydrogen is 130–170 mL). The parameters of the three-center kinetic model were evaluated in three stages. At the first stage, k_ij_, k_pj_, and k_dj_ were affecting the polymerization rate, are assessed by solving the problem of minimizing the sum of the least squares of deviations of the measured polymerization rate values and calculated using the kinetic model. At the second stage, the rate constants of the chain transfer reactions, which mainly affect the molecular weight, are found by solving the problem of minimizing the sum of the least squares of the experimental deviations, $M_{n,e}$ and $M_{w,e}$, and the corresponding values calculated using the kinetic model. At the third stage, the optimization was performed by combining the objective functions of the first and second stages, and the results of the two stages were used as the initial data set for the third stage. For the chain growth coefficients on three AC types, the values $E_{p1}/R$ = 21,200, $E_{p2}/R$ = 741.73, and $E_{p3}/R$ = 2050 were obtained, while for the chain-to-monomer-transfer coefficients are $E_{m1}/R$ = 514.32, $E_{m2}/R$ = 2180, and $E_{m3}/R$ = 1960. It should be noted that the results of decomposing the experimental MWD polymer samples into Flory fractions were only used in [18] to find the number of AC types. However, unlike the case in the present study, the results were not used at the second stage to identify the kinetic coefficients of the reactions.

## 4. Conclusions

A polycenteric model of the kinetics of polymerizing propylene based on the TiCl_4_/DBP/MgCl_2_ + TEA/CHMDMS catalytic system is compiled, reflecting the temperature dependence of the kinetic parameters. Using the experimental PP MWD deconvolution, the minimum number of the active center types of the catalytic system was found, which is important for a suitable description of the molecular weight distribution of the synthesized polymer (Ns = 5). The model includes five AC types of AC and, respectively, five coefficients of activation for each of the growth, chain transfer to hydrogen, monomer, and AC deactivation. A three-stage algorithm is proposed for identifying the activation energies, and a preexponent was developed for the Arrhenius dependences of each coefficient on the polymerization temperature. The model is implemented in the Mathematica package. An algorithm was created for a step-by-step identification of the key kinetic parameters. The proposed kinetic model, together with the identified parameters, describes consistently the experimental yields of polypropylene and its molecular weight characteristics when varying the initial concentrations of the monomer and hydrogen within the ranges studied. The simulation results acceptably fit with the experimental data and show that the identified values of the kinetic parameters allow for a proper predicting of the polymer yield and its molecular mass characteristics within the polymerization temperature range of T = 40–90° C. The obtained values of the kinetic parameters can be used in developing a non-isothermal model.

## Figures and Tables

**Figure 1 polymers-14-05183-f001:**
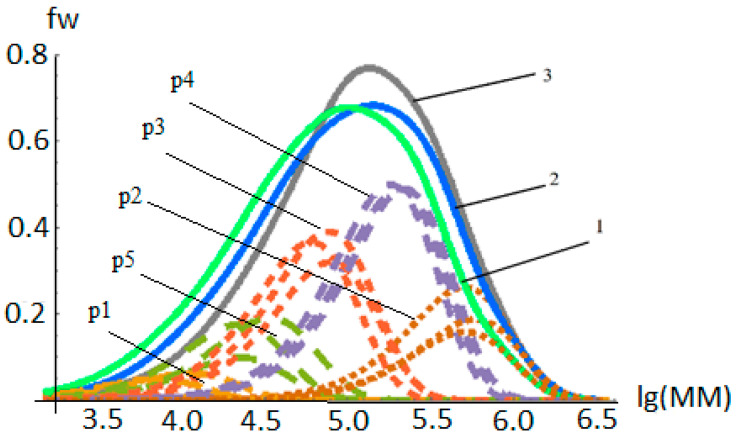
Comparison of MWD deconvolutions of PP samples obtained at polymerization temperatures: T = 40 °C (1), T = 50 °C (2), T = 90 °C (3) (polymerization conditions in Table 2). Semi-solid curves show the distribution of polymer fractions formed on isolated active centers (Table 3).

**Figure 2 polymers-14-05183-f002:**
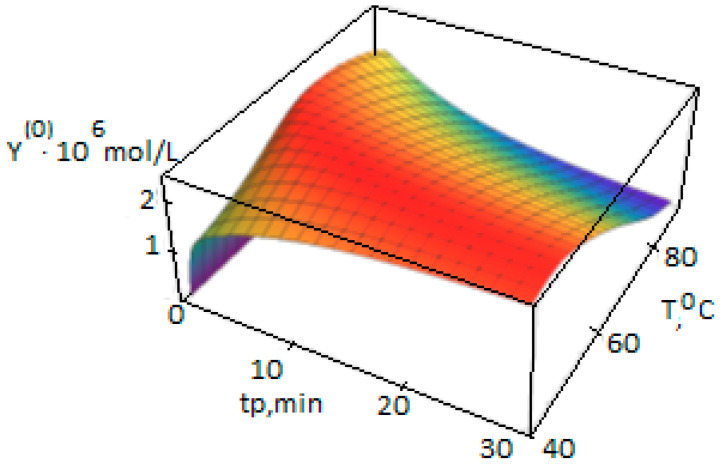
Dependence of the AC concentration $Y^{\left(0\right)}$, mol/L, on the temperature T and the polymerization duration t_p_, calculated according to the polycenter model using the growth, activation, and deactivation energies from Table 3 (polymerization conditions in Table 1).

**Figure 3 polymers-14-05183-f003:**
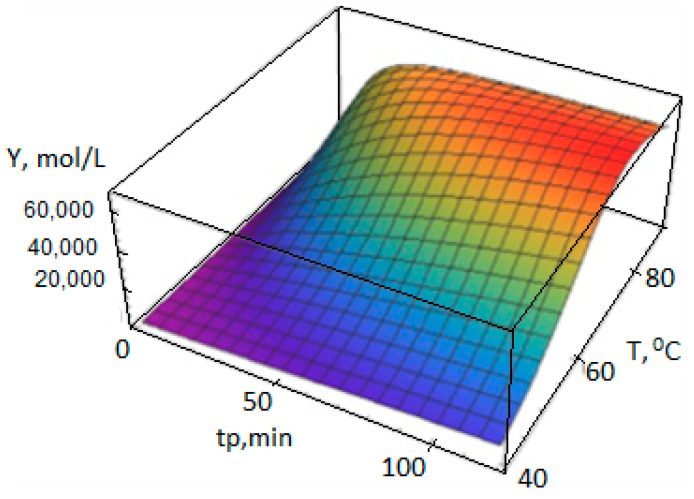
Dependence of the output of PP Y_p_, g_PP_/g_cat_, on the temperature T and the polymerization duration t_p_, calculated according to the polycenter model using the growth, activation, and deactivation energies from Table 3 (polymerization conditions in Table 1).

**Figure 4 polymers-14-05183-f004:**
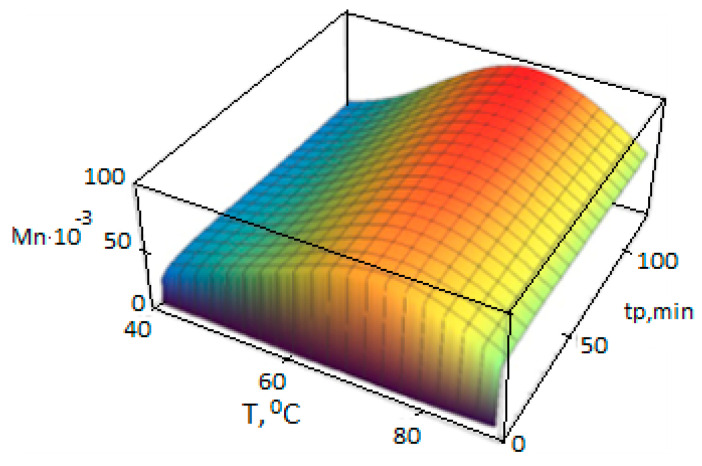
Dependence of the calculated average molecular weight of PP on the temperature T and the polymerization duration t_p_, calculated using the polycenteric model using the activation energies from Table 3 (polymerization conditions in Table 2).

**Figure 5 polymers-14-05183-f005:**
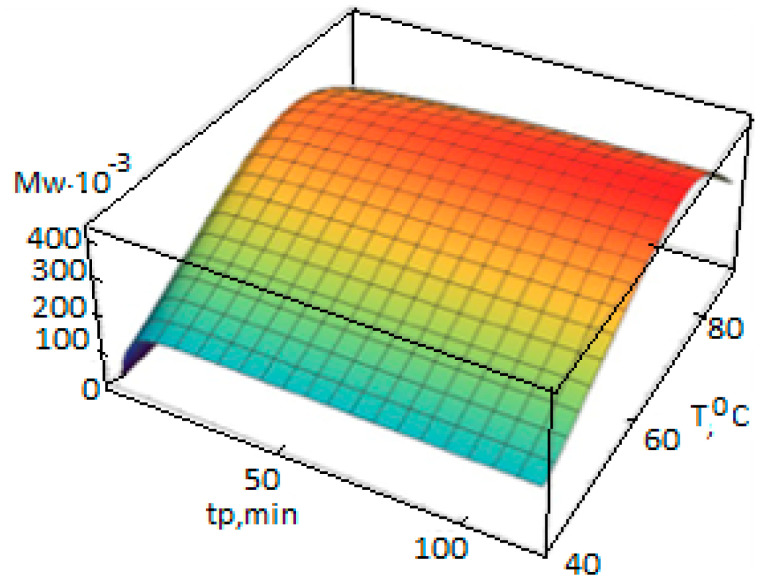
Dependence of the average mass molecular weight of PP on the temperature T and the polymerization duration t_p_, calculated according to the polycenteric model using the activation energies from Table 3 (polymerization conditions in Table 2).

**Figure 6 polymers-14-05183-f006:**
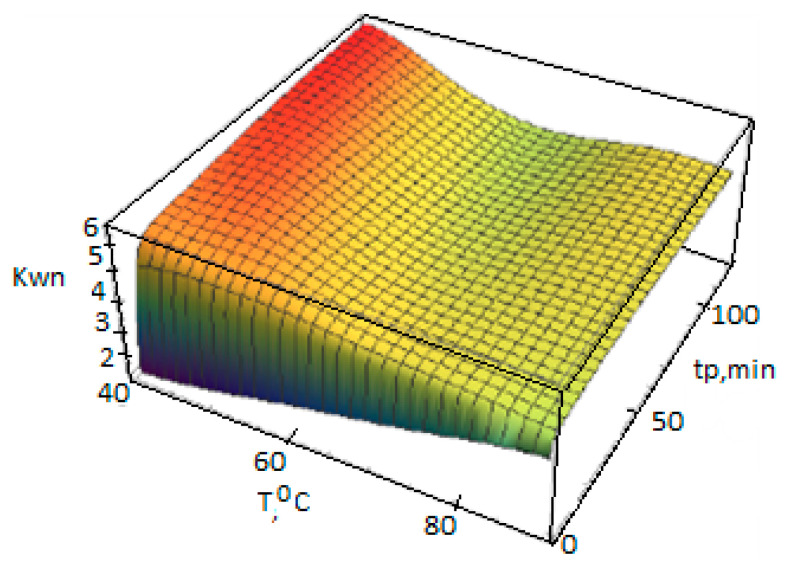
Dependence of the polydispersity coefficient of PP on the temperature T and the polymerization duration t_p_, calculated according to the polycenteric model using the activation energies from Table 3 (polymerization conditions in Table 2).

**Figure 7 polymers-14-05183-f007:**
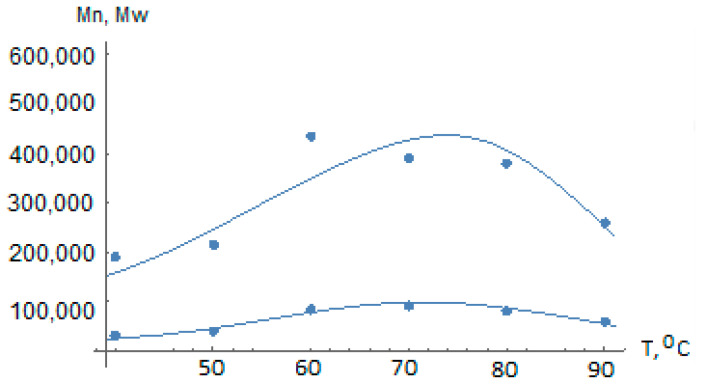
Comparison of the dependences $M_{n}\left(120,T\right)$ and $M_{w}\left(120,T\right)$ calculated by the polycentric model with the activation energies from Table 3, and the experimental data from Table 1 (relative error Δn_40_ = 0.196, Δn_50_ = 0.156, Δn_60_ = 0.067, Δn_70_ = 0.079, Δn_80_ = 0.075, Δn_90_ = 0.071; Δw_40_ = 0.167, Δw_50_ = 0.139, Δw_60_ = 0.195, Δw_70_ = 0.095, Δw_80_ = 0.070, Δw_90_ = 0.027).

**Figure 8 polymers-14-05183-f008:**
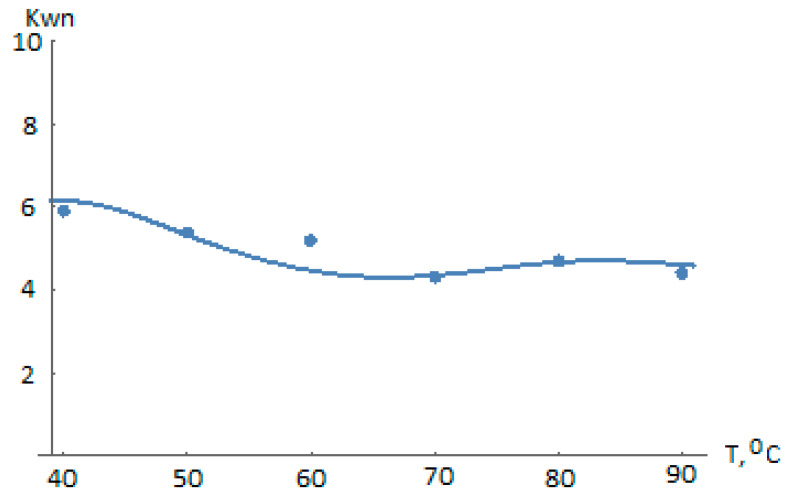
Comparison of the dependence K_wn_ (120, T) calculated using the polycenteric model (activation energies from Table 3 and experimental data from Table 1; relative errors Δ_40_ = 0.042, Δ_50_ = 0.014, Δ_60_ = 0.141, Δ_70_ = 0.011, Δ_80_ = 0.006, Δ_90_ = 0.048).

**Table 1 polymers-14-05183-t001:** Kinetic scheme of the liquid-phase polymerization of propylene.

Reaction Type	Reaction
Activation	$C_{j}+A\overset{k_{aj}}{\to }P_{j,0}$, $j=\bar{1,N_{s}}$
Initiation	$C_{j}+M\overset{k_{ij}}{\to }P_{j,1}$, $j=\bar{1,N_{s}}$
Chain growth	$P_{j,r}+M\overset{k_{pj}}{\to }P_{j,r+1}$, $j=\bar{1,N_{s}}$, r = 1, 2…
Chain transfer to hydrogen	$P_{j,r}+H_{2}\overset{k_{\mathrm{Hj}}}{\to }P_{j,0}+D_{j,r}$, $j=\bar{1,N_{s}}$, r = 1, 2…
Chain transfer to monomer	$P_{j,r}+M\overset{k_{\mathrm{mj}}}{\to }P_{j,0}+D_{j,r}$, $j=\bar{1,N_{s}}$, r = 1, 2…
Deactivation	$P_{j,0}\overset{k_{\mathrm{dj}}}{\to }C_{d}$, $P_{j,r}\overset{k_{\mathrm{dj}}}{\to }C_{d}+D_{j,r}$, $j=\bar{1,N_{s}}$, r = 1, 2…

**Table 2 polymers-14-05183-t002:** Polymerization temperature effect on the experimental outputs and average properties of PP synthesized on catalyst system TiCl_4_/DBP/MgCl_2_ + TEA/CHMDMS ($t_{p}$ = 120 min, P = 30–32 kg/cm^2^, m_H2_ = 0.15 mol, m_cat_ = 0.015 g of catalyst, f_Ti_ = 0.025 mass., Al/Ti = 1500 mol, and Al/Si = 20 mol [1].

T, °C	${\tilde{Y}}_{p,e},\;{\mathrm{kg}}_{\mathrm{pp}}/g_{\mathrm{cat}}$	M_n,e_(×10^−3^), g/mol	M_w,e_(×10^−3^), g/mol	$K_{wn}$
40	10	32	190	5.9
50	19	40	216	5.4
60	47	84	435	5.2
70	57	91	390	4.3
80	60	81	380	4.7
90	58	59	260	4.4

**Table 3 polymers-14-05183-t003:** Mass fractions and average calculated molecular weights of polymer fractions determined by the MWD deconvolution of PP samples synthesized at the temperatures of T = 40–90 °C (polymerization conditions in Table 2).

T, °C	p1 Mn1	p2 Mn2	p3 Mn3	p4 Mn4	p5 Mn5
40	0.039 3549.24	0.141 11,200.3	0.293 30,234.5	0.400 90,494.9	0.127 253,306
50	0.050 5739.31	0.152 17,071.2	0.256 39,726	0.392 110,026	0.150 290,661
60	0.025 9445.41	0.141 32,145.3	0.277 84,195.1	0.386 24,0571	0.171 628,001
70	0.014 12,852.0	0.067 20,043.3	0.317 82,935.2	0.464 266,407	0.138 778,857
80	0.035 9993.72	0.106 32,740.4	0.322 82,795.6	0.414 229,880	0.123 642,568
90	0.006 2758.44	0.079 11,268.2	0.313 38,394.9	0.392 101,476	0.210 252,532

**Table 4 polymers-14-05183-t004:** Calculated energies of chain growth, activation and deactivation of active centers, chain transfer to monomer and hydrogen.

E, cal/mol	AC1	AC2	AC3	AC4	AC5
$E_{pj}$	13,587.8	16,985.6	22,380.4	21,691.6	20,411.3
$E_{aj}$	36,936.7	2181.47	24,193.3	22,359.6	2181.47
$E_{dj}$	12,536.9	15,890.4	22,460.1	23,426.5	17,764.4
$E_{mj}$	2769.28	10,294.8	30,433.9	14,867.9	11,642.8
$E_{hj}$	38,146.4	36,261.6	3575.38	36,921.1	47,616.1

## Data Availability

The data presented in this study are available on request from the corresponding author.

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
