# Peer review of "Temperature Dependence of the Kinetic Parameters of the Titanium–Magnesium Catalyzed Propylene Polymerization"

_polymers, 2022, doi:10.3390/polym14235183_

Round 1

Reviewer 1 Report

This manuscript looks interesting and useful. However, the following comments need to deal.

Comments:

1) On lines 240-241 of page 7,  It is better to write Yang et al [11]....instead of "The authors of [11] came to the conclusion that the ideal mixing model (CSTR) can be used to describe the flow structure in a loop reactor of propylene polymerization at a recirculation coefficient above 50."

2) Correction is needed in English language throughout the manuscript. 

3) There should be indication of Figure 1 on the semi-solid curves as like solid curves and more discussion of Figure 1 is needed.

4) On line 297 of page 9,  There is "Table 3. Calculated energies of chain growth, activation and deactivation of active centres, chain transfer to monomer and hydrogen" and on line 299 of page 9, there is also "Table 3. Calculated energies of chain growth, activation and deactivation of active centres, chain transfer to monomer and hydrogen". Such way of writing is necessary to correct. 

5) Figures 4, 5 and 6  should also mention in the text of the manuscript  which  are missing. 

6) Elaborate the conclusions by including the exact findings from  results and discussion and correct "T=40-90o° C" as T=40-90 °C".

7) Rewrite the Acknowledgments in the proper way by mentioning the country name of funding agency. 

Author Response

1) We corrected. Please see page 8, line 653.

2) See the article.

3) We added discussion of Figure 1 (page 9, lines 821-827), and changed the figure 1 (page 9, line 817). 

4) We corrected. Please see page 9, lines 828-830.

5) We mention Figures 4, 5 and 6. Please see page 11, line 945.

6) We corrected throughout the article.

7) We rewrote the Acknowledgments (page 13, lines 1087-1088). 

Reviewer 2 Report

In this manuscript, Bronskaya et.al developed a polycentric kinetic model for the polymerization of polypropylene catalyzed by the titanium-magnesium catalyst. The authors can obtain the kinetic parameters using a three-stage algorithm. The authors started with a kinetic model for a single active center. Then, the authors identified 5 active centers for the polymerization of propylene by deconvoluting the molecular weight distribution. After that, the kinetics parameters were obtained for each active center.  This work shows some potentials in predicting the reaction of industrial synthesis of polypropylene, I would recommend it for publication after major revision. Here are several comments below that would be helpful:

1.      In line 13, “the Ziegler-Natt type” should be “the Ziegler-Natta type”.

2.      I am confused why this kinetic model is specific for titanium-magnesium catalysts. Does this model also work for other polycentric heterogeneous catalysts with similar mechanisms?

3.      In lines 17-18, “When developing polycentric kinetic models, there is a difficulty associated with the determination of the values of the kinetic constants of the rates of elementary reactions (stages) of the polymerization process. ”. This sentence is almost unreadable with 5 “of”. The authors should rephrase it.

4.      The language throughout the paper needs to be polished.

5.      In line 22, what is “velocity constant”?

6.      In line 93, what is “policentro”? Do the authors mean polycentric?

7.      In line 95, the authors should explain what the catalyst (TiCl4/DBP/MgCl2+TEA/CHMDMS) consists of as there are many abbreviations used without explanation.

8.      In the last row of table 1, it is confusing that Pj,0 can form only Cd, while it can also form both Cd and Dj,r. Did the authors mean Pj,r → Cd+Dj,r?

9.      In line 117, “j – type center” should be “j – center type” or “j- type of center”

10.   In line 119, how did the authors define chain length r? The authors need to clarify it in the text.

11.   The authors need to reformat their equations.

12.   The authors should use consistent units, for example, do not switch back and forth between kJ/mol and cal/mol. In addition, the authors should use the correct units, for example, “mol/l” should be “mol/L”.

13.   There are too many typos throughout the paper.

14.   The title of table 3 was repeated.

15.  The authors should use English units in figure 2.

Author Response

  1. We corrected throughout the article.
  2. This model work for other polycentric heterogeneous catalysts.  When considering catalysts, it is possible both complication and simplification of the model.
  3. Please see page 1, lines 19-20.
  4. We corrected throughout the article.
  5. Please see page 1, line 23.
  6. We corrected throughout the article
  7. We corrected it. Please see page 3, line 245.
  8.  Pj,r → Cd+Dj,r is the deactivation reaction that leads the the deactivation of AC catalyst and deactivation of the active centres of the polymer chain.
  9. We corrected it. Please see page 4, line 370.
  10. We corrected it. Please see page 4, line 372.
  11. What format should the equations be in?
  12. We corrected units throughout the article.
  13. We corrected throughout the article
  14. We corrected it. Please see page 10, line 907.
  15. Please see page 10, line 911.

Round 2

Reviewer 1 Report

The revised manuscript looks fine. 

Reviewer 2 Report

The overall language of the paper has been improved, and the authors have addressed most of the comments. I would recommend it for publication after minor revision.

1. There is a typo in the title. "Ptopylene" should be propylene. In addition, there are still some typos throughout the paper.

2. The authors should not switch back and forth between "center" and "centre".

3. In the last row of Table 1, I am still confused about the meaning of Pj,0 → Cd + Dj,r. How does an active monomer center form a dead polymer chain? There is no polymer on the left of this equation, so where does the dead polymer come from?

4. The equations can be formatted in a better way so that they are easier to read. For example, in eq. 8, the content in the round brackets could be put in the same row. Same for eq. 12. The equations are not aligned well. 

5. In line 234, ΔEef should be ΔEef.

6. The author didn't correct the unit throughout the paper, for example, the unit of min in figures 3-6 is still in Russian.

Author Response

  1. We corrected. Please see page 1, line 2.

2. We corrected. Please see throughout the article.

3. It was a typo. Thanks for your comment, we really appreciate it.

We corrected. Please see Table 1.

4. We corrected. Please see eq. 8, eq. 12, eq. 26, eq. 28.

5. We corrected. Please see page 7, line 234.

6. We corrected. Please see figures 3-6.
